# Efficacy of Neoadjuvant Hypofractionated Chemoradiotherapy in Elderly Patients with Locally Advanced Rectal Cancer: A Single-Center Retrospective Analysis

**DOI:** 10.3390/cancers16244280

**Published:** 2024-12-23

**Authors:** Jae Seung Kim, Jaram Lee, Hyeung-min Park, Soo Young Lee, Chang Hyun Kim, Hyeong Rok Kim

**Affiliations:** Department of Surgery, Chonnam National University Hwasun Hospital and Medical School, Hwasun 58128, Republic of Korea; cnumed12@gmail.com (J.S.K.); ramraming@naver.com (J.L.); smart1015423@gmail.com (H.-m.P.); cksantiago8@gmail.com (C.H.K.); drkhr@jnu.ac.kr (H.R.K.)

**Keywords:** hypofractionated chemoradiotherapy, elderly, rectal cancer, neoadjuvant

## Abstract

The application of long-course chemoradiotherapy (LCRT) in elderly patients can be challenging due to increased risks of complications. This study aimed to assess the efficacy of neoadjuvant hypofractionated chemoradiotherapy (HCRT) in elderly patients with locally advanced mid-to-low locally advanced rectal cancer. Patients were categorized into three groups based on their treatment strategies: neoadjuvant HCRT, neoadjuvant LCRT, and upfront surgery. Three-year relapse-free survival, three-year local recurrence-free survival, and five-year overall survival showed no significant difference among the three groups. Multivariate analysis also showed that the treatment strategy was not associated with survival outcomes. Neoadjuvant HCRT demonstrated reduced radiotherapy-related complications and acceptable long-term oncologic outcomes. Therefore, neoadjuvant HCRT may be considered as a viable alternative for elderly patients with locally advanced mid-to-low locally advanced rectal cancer.

## 1. Introduction

Colorectal cancer (CRC) is the third most common cancer worldwide and the second leading cause of cancer-related mortality [1]. While CRC incidence rates are the highest in European regions, it is notable that the rectal cancer incidence rates in Eastern Asia rank among the highest worldwide [2]. The current standard approach for locally advanced rectal cancer (LARC) is total neoadjuvant therapy (TNT) followed by radical surgery [3,4]. However, before the advent of TNT, neoadjuvant long-course chemoradiotherapy (LCRT) had been the standard treatment for LARC for several decades [5].

LCRT offers advantages such as enhancing the tumor response and enabling more effective surgical outcomes [5,6]. However, LCRT is associated with a higher risk of adverse effects than other treatment strategies, making its use challenging, especially in elderly patients who often present with comorbidities and have an increased susceptibility to treatment-related complications [7,8]. Therefore, in elderly patients with LARC, compliance with LCRT has been reported to be low [7]. Moreover, applying TNT in this population is even more challenging. This suggests the need for a tailored approach in elderly patients with LARC. Alternative options for treating LARC include upfront surgery and short-course radiotherapy [9,10]. Upfront surgery offers a quick resolution of the primary tumor, avoids radiation-related toxicity, and reduces medical costs but may miss the opportunity for tumor downstaging and increase the risk of local recurrence [11]. Recent reports have demonstrated that short-course radiotherapy with a delayed surgery is a viable alternative [12,13,14]. In addition, some authors have suggested that hypofractionated chemoradiotherapy (HCRT) can be another option for the treatment of LARC [15]. However, there is limited research on the feasibility of HCRT in elderly patients.

Therefore, the present study was designed to evaluate the efficacy and feasibility of neoadjuvant HCRT in elderly patients with LARC by comparing the clinicopathological characteristics, short-term outcomes, and long-term survival outcomes of neoadjuvant HCRT, LCRT, and upfront surgery.

## 2. Materials and Methods

### 2.1. Patients

This study was approved by the Institutional Review Board of Chonnam National University Hwasun Hospital (IRB No. CNUHH-2024-071), with informed consent waived due to the study’s low risk and the use of deidentified patient data. We performed a retrospective analysis of patients diagnosed with LARC between January 2013 and December 2020. The baseline characteristics of these patients and their clinicopathological information were obtained from a prospectively collected clinical database, which contains the records of patients who have undergone surgery within a specified period. The inclusion criteria were as follows: primary mid-to-low rectal cancer, age of 70 years or older, and clinical stage II or III based on pretreatment rectal magnetic resonance imaging (MRI) or abdominopelvic computed tomography (CT). Patients were excluded if they had upper rectal cancer, clinical stage I or IV disease, or recurrent rectal cancer.

Patients underwent pretreatment assessments, such as laboratory tests including serum carcinoembryonic antigen (CEA) level measurement, colonoscopy, abdominopelvic and chest CT, rectal MRI, and positron emission tomography (PET). When a rectal MRI was unavailable, an abdominopelvic CT was used as a substitute for local staging. Low rectal cancer was defined as tumors located less than 6 cm from the anal verge, while mid rectal cancer was defined as tumors located 6–10 cm from the anal verge, both based on rectal MRI or rigid sigmoidoscopy. Metastases were assessed using abdominopelvic CT and/or PET.

### 2.2. Neoadjuvant Treatment

Patients were allocated to different treatment strategies based on the attending surgeon’s and oncologist’s discretion, considering factors such as the patient’s overall condition, comorbidities, age, tumor location, and logistical challenges, including distance from the treatment center. However, there was no standardized criteria for treatment allocation. Radiotherapy was performed in an outpatient setting utilizing CT-based three-dimensional field treatment planning. Patients receiving HCRT were treated with either 33 Gy in 10 fractions to the whole pelvis with intensity-modulated radiotherapy or 35 Gy in 10 fractions to the primary bulky tumor via simultaneous integrated boost and 33 Gy to the remaining pelvis with intensity-modulated radiotherapy. Those undergoing LCRT received 48.6 to 50.4 Gy in 25 to 28 fractions. For most patients, concurrent oral capecitabine was administered at a dose of 1650 mg/m^2^/day 5 days/week for up to 10 days during radiotherapy. Others were treated with intravenous 5-fluorouracil and leucovorin. Upon completion of radiotherapy, patients were restaged using either rectal MRI or abdominopelvic CT. Patients who received radiotherapy were assessed for complications throughout the treatment course. Various types of complications, including nausea/vomiting, headache, bowel habit changes, anorexia, dermatitis, dysuria, anal pain, abdominal pain, general weakness, and leg edema were recorded, and the total number of complications was documented. If a patient experienced multiple complications, each instance was counted separately. Toxicity was graded using the Common Terminology Criteria for Adverse Events (CTCAE), Version 5.0 [16].

### 2.3. Surgery and Surveillance

Surgery was scheduled to take place 6 to 10 weeks after the completion of chemoradiotherapy. The types of surgeries included low anterior resection, intersphincteric resection, abdominoperineal resection, and transanal excision. The decision to perform radical surgery or local excision was made at the discretion of the attending surgeon. Radical surgery was performed following the principles of total mesorectal excision for optimal oncological outcomes [17,18]. A diverting stoma was created if there was a high risk of anastomotic leakage, particularly in patients who had undergone prior chemoradiotherapy. The pathological stage was determined in accordance with the 8th Edition of the American Joint Committee on Cancer’s *AJCC Cancer Staging Manual* [19]. Based on the patient’s pathological stage or performance status, 5-fluorouracil-based adjuvant chemotherapy was administered. The postoperative follow-up involved outpatient visits; the first visit was scheduled two weeks after discharge, and subsequent visits were planned every six months. Follow-up assessments including serum CEA testing and chest and abdominopelvic CT were performed semiannually, and a colonoscopy was performed annually or biennially. A PET/CT was also performed when recurrence was suspected. Recurrence was determined through clinical and radiological evaluations or was confirmed histologically. Local recurrence was defined as recurrence within the pelvic cavity or at the anastomotic site without distant metastases.

### 2.4. Statistical Analysis

Patient characteristics, clinicopathological data, radiotherapy-related complications, and survival outcomes were compared. Relapse-free survival (RFS) was defined as the period from the date of surgery to the date of the first documented recurrence. Local recurrence-free survival (LRFS) was defined as the period from the date of surgery to the date of local recurrence or death. Distant metastasis-free survival (DMFS) was defined as the period from the date of surgery to the date of distal metastasis detection or death. Overall survival (OS) was defined as the period from the initiation of treatment, whether radiotherapy or surgery, to the date of death from any cause. Nominal and categorical data were analyzed using the χ^2^ test or Fisher’s exact test, and continuous variables were compared using Student’s t-test. Survival analysis was conducted using the Kaplan–Meier method and log-rank test. Multivariate analysis was performed on variables that showed significant differences in the univariate analysis, as well as on the treatment strategy in which we were interested, using a Cox proportional hazards regression model. All statistical analyses were conducted using SPSS software (version 25.0; IBM Inc., Armonk, NY, USA) with a significance threshold of *p* < 0.05.

## 3. Results

We reviewed a total of 660 patients aged 70 years or older with LARC. After applying the exclusion criteria, 364 patients were excluded, resulting in a final cohort of 296 patients. Of these, 30 (10.1%) patients received HCRT, 195 (65.9%) underwent standard LCRT, and 71 (24.0%) underwent upfront surgery (Figure 1). In the HCRT group, 28 (93.3%) patients underwent radical surgery, and 2 (6.7%) underwent local excision. In the LCRT group, 188 (96.4%) patients proceeded to radical surgery following neoadjuvant therapy, while 7 (3.6%) patients underwent local excision (Figure 1). A post hoc power analysis was conducted to evaluate the statistical power of the study. The analysis revealed that the calculated statistical power for RFS, LRFS, and OS was 0.27, 0.24, and 0.14, respectively. To achieve a statistical power of 80% (α = 0.05), the estimated total sample sizes required were 550 for RFS, 346 for LRFS, and 1867 for OS.

The baseline characteristics of the patients are presented in Table 1. The HCRT group had a higher proportion of patients with an American Society of Anesthesiologists (ASA) score of 3 or 4 (43.3%) than the LCRT (16.9%) or upfront surgery (15.5%) groups (*p* = 0.002). Significantly more patients in the upfront surgery group (85.9%) than in the LCRT (54.9%) or HCRT (60.0%) groups had mid rectal cancer (*p* < 0.001).

Patients who underwent LCRT experienced a significantly higher incidence of complications (48.7%) than those in the HCRT group (16.7%) (*p* = 0.001) (Table 2). In the LCRT group, the most commonly reported complications were anal pain, bowel habit changes, and anorexia, which were also the predominant complications observed in the HCRT group. CTCAE grade 3 or 4 events were rare; most cases were classified as CTCAE grade 1 or 2. The LCRT group had a higher incidence of grade 1 and grade 2 complications than the HCRT group (38.9% vs. 13.3% and 8.2% vs. 3.3%, respectively). However, these differences were not statistically significant (*p* = 0.911).

Clinical T downstaging by chemoradiotherapy assessed by rectal MRI was more frequently observed in the LCRT group (n = 100, 51.3%) than in the HCRT group (n = 11, 36.7%); however, this difference was not statistically significant (*p* = 0.136). Similarly, N downstaging was more frequently observed in the LCRT group than in the HCRT group (34.9% vs. 16.7%, p = 0.047). The pathological complete response (pCR) rate was significantly lower in the HCRT group than in the LCRT group (*p* = 0.002) (Table 3). The proportion of patients with pathological T4 tended to be higher in the HCRT group (13.3%) than in the LCRT group (5.6%), while the proportion of patients with pathological T0 or 1 tended to be lower in the HCRT group (10.0%) than in the LCRT group (21.5%). However, these differences were not statistically significant (*p* = 0.243). Similarly, more patients in the HCRT group tended to have pathological N2 (17.9%) than those in the LCRT group (5.9%) (*p* = 0.059).

Table 4 presents the result of the univariate survival analysis of the entire cohort. Several variables, including hemoglobin levels, tumor location, tumor differentiation, clinical N stage, lymphovascular invasion, perineural invasion, and tumor regression grade, were associated with survival outcomes. However, the treatment strategy did not show any significant differences in survival outcomes (3-year RFS, HCRT 83.0% vs. LCRT 77.2% vs. upfront surgery 83.2%, *p* = 0.411; 3-year LFRS, 93.1% vs. 93.2% vs. 93.5%, *p* = 0.464; 3-year DMFS, 85.8% vs. 79.9% vs. 84.8%, *p* = 0.552; 5-year OS, 65.1% vs. 67.0% vs. 67.7%, *p* = 0.682) (Figure 2). The multivariate analysis also indicated that the treatment strategy was not associated with survival outcomes (Table 5).

## 4. Discussion

The aim of this study was to evaluate the efficacy and feasibility of neoadjuvant HCRT by comparing the clinicopathological characteristics, short-term outcomes, and long-term survival outcomes of neoadjuvant HCRT, LCRT, and upfront surgery in elderly patients with mid-to-low LARC. The HCRT group included more patients with a high ASA score than the other groups. The HCRT group also had a lower incidence of radiotherapy-related complications than the LCRT group. Although HCRT was associated with relatively lower rates of tumor downstaging and pCR, it showed comparable oncologic outcomes to conventional LCRT.

Over the past few decades, following the German trial, LCRT followed by surgery and adjuvant chemotherapy have become the standard treatment for rectal cancer [20]. LCRT prolongs the duration of preoperative therapy, potentially improving tumor response and facilitating better surgical outcomes; however, it also increases the risk of treatment-related side effects [21]. Alternative treatment options exist, such as upfront surgery, which involves immediate tumor resection followed by adjuvant therapies as necessary. This approach allows for a quick resolution of the primary tumor, reduces radiation-related toxicity, and lowers medical costs; however, it may lead to a missed opportunity for tumor downstaging and increased sphincter preservation [11]. Short-course radiotherapy, which involves a shorter period of radiation before surgery, has the advantage of a reduced treatment period, which can potentially increase patient compliance, while aiming to reduce the tumor size and improve surgical outcomes [12,13,14]. However, delivering a high dose of radiation in a short period may increase the risk of complications, and the brief treatment duration may not be sufficient to achieve tumor downstaging [22].

Elderly patients are often vulnerable to treatment-related complications due to comorbidities, reduced functional status, and poor treatment compliance, making the application of neoadjuvant therapy more challenging [7,23,24]. These issues have led to emphasis on the need for alternative therapeutic approaches that are better suited to the health status of elderly patients, particularly in cases of rectal cancer [25]. A recent study proposed a new standard of care for patients aged 75 years or older with LARC, demonstrating that a treatment regimen of short-course radiotherapy followed by delayed surgery may represent a preferred therapeutic approach [26]. In this study, we combined the advantages and limitations of various preoperative neoadjuvant therapies and applied HCRT to elderly patients. Our goal was to assess the efficacy and feasibility of HCRT, aiming to improve treatment compliance while achieving oncologic outcomes comparable to LCRT.

In this study, various complications such as anal pain and bowel habit changes were observed following LCRT. These findings are consistent with those reported in previous studies [27,28]. Anal pain is attributed to radiation-induced proctitis occurring as a result of radiation therapy for various pelvic malignancies [29,30]. Changes in bowel habits, manifesting as either diarrhea or constipation, are caused by pelvic dysfunction following treatment [31]. Ultimately, these issues contribute to decreased compliance with chemoradiotherapy. Our findings indicate that HCRT resulted in significantly fewer complications than LCRT. This suggests that HCRT may help mitigate issues related to poor compliance, particularly in elderly patients.

On the other hand, HCRT demonstrated inferior results in terms of tumor downstaging and the pCR rate compared with LCRT. In particular, the rate of downstaging observed via MRI was lower in the HCRT group than in the LCRT group (clinical T stage, 36.7% vs. 51.3%; clinical N stage, 16.7% vs. 34.9%), and the pCR rate was also reduced (10.0% vs. 15.4%). Notably, there was a higher tendency for patients in the HCRT group to present with pathological N2 stage disease. This is likely due to the shorter duration of CRT, which may compromise local control. Nonetheless, a previous study with a larger sample size has reported comparable pCR rates [14]. This indicates that the limited patient population in our study should not lead to a definitive conclusion regarding the local control efficacy of HCRT. Moreover, although the pCR rate was lower, it did not significantly impact local recurrence or survival outcomes. There was no significant difference between the LCRT and HCRT groups in terms of RFS, LRFS, DMFS, or OS. Therefore, HCRT may serve as a viable alternative for elderly patients, offering reduced complications and comparable long-term outcomes compared with LCRT. By effectively balancing oncologic efficacy with reduced toxicity, HCRT is particularly suitable for elderly patients who may not be ideal candidates for LCRT or TNT. Furthermore, the reduced number of radiotherapy sessions associated with HCRT has the potential to enhance patient compliance and improve accessibility to treatment, especially in resource-limited settings.

The lower pCR rate observed with HCRT may limit the applicability of organ-preserving strategies, such as local excision or the watch-and-wait approach. This study focused on patients who underwent surgery; therefore, a watch-and-wait strategy was not included. Furthermore, there was a notable difference in local excision rates. For elderly patients, less extensive surgery or the watch-and-wait strategy is important for recovery and survival, emphasizing the importance of selecting a treatment that can achieve higher pCR rates. Although the pCR rate in the HCRT group was lower in this study, the previous literature has reported that HCRT and LCRT show comparable pCR rates in larger cohorts [14]. Further research is needed to assess the feasibility of organ-preserving strategies following HCRT.

While HCRT did not show significant differences in survival outcomes compared with LCRT, this study also found no difference when compared with upfront surgery. LCRT has been reported to have superior oncologic outcomes compared with upfront surgery in several studies [21]. However, our study did not reveal this disparity, which may be attributed to differences in baseline characteristics. The upfront surgery group included more patients with mid rectal cancer and lower CEA levels. This suggests that patients in the upfront surgery group may have been more suitable candidates for surgery or more likely to achieve favorable long-term outcomes than those who received chemoradiotherapy. In fact, tumor location was an independent prognostic factor for patients with rectal cancer that affected survival outcomes [32], suggesting that the higher proportion of mid rectal cancer cases in the upfront surgery group may have mitigated the survival differences between the upfront surgery and LCRT groups.

The current standard treatment of rectal cancer is TNT [3]. It has been established as the standard since 2020 [5]. Compared with conventional CRT, TNT offers several advantages, including improved survival outcomes and an increased likelihood of implementing organ-preserving strategies [33,34]. However, this study was conducted before the widespread adoption of TNT; thus, patients receiving this treatment were not included. In addition, TNT was not covered by national health insurance in Korea during the study period, which limits its applicability. Research on the feasibility of TNT in elderly patients remains limited. As mentioned, elderly patients have a poor performance status, making it difficult to apply standard approaches. The rates of adjuvant chemotherapy or LCRT in this population have been notably low; thus, the optimal treatment strategy for elderly patients remains mixed, and there are significant limitations that are inherent to retrospective and population-based analyses [35]. This indicates that achieving favorable results from applying TNT in elderly patients may be challenging.

There are a few limitations to this study. Like many other retrospective studies, there may be biases in patient selection, as patients were allocated to each treatment strategy at the attending surgeon’s and oncologist’s discretion. Consequently, the HCRT group contained more patients with higher ASA scores, implying that HCRT may have been selected for patients who were less likely to comply with LCRT. Statistical methods such as propensity score matching could help to minimize these discrepancies and make the groups more comparable. In fact, the lack of standardized criteria for treatment allocation reflects the real-world decision-making process, where multiple factors, including patient preference, logistical challenges, and clinical judgment, play a role. While this approach aligns with practical clinical settings, it introduces a variability that may limit the generalizability of the findings. Second, due to the small sample size and low event rates, the post hoc power analysis demonstrated a low statistical power. This suggests that a type II error may have occurred, where a true difference among treatment strategies exists but could not be detected. Third, due to the limitations inherent to the retrospective design of this study, we were unable to obtain sufficient data on biomarkers such as KRAS, NRAS, BRAF, and microsatellite instability to perform a meaningful analysis or incorporate these into the results. In addition, the retrospective design of this study may have limited the completeness of complication data collection. Finally, patients who achieved a clinical complete response following neoadjuvant chemoradiotherapy could have been candidates for the watch-and-wait strategy, but this approach was not included for the present study. Nevertheless, this study holds significance as one of the very few to demonstrate that neoadjuvant HCRT could be a viable treatment strategy for elderly patients. Further studies with larger patient cohorts are needed to more definitively validate the findings of this research. Although our study is limited to rectal cancer, we believe that our findings can also provide valuable insights into the treatment of elderly patients with other cancer types eligible for radiation therapy [36].

## 5. Conclusions

Neoadjuvant HCRT was associated with a relatively low incidence of radiotherapy-related complications and acceptable long-term oncologic outcomes in elderly patients with LARC. Therefore, neoadjuvant HCRT may be considered as a viable alternative for these patients.

## Figures and Tables

**Figure 1 cancers-16-04280-f001:**
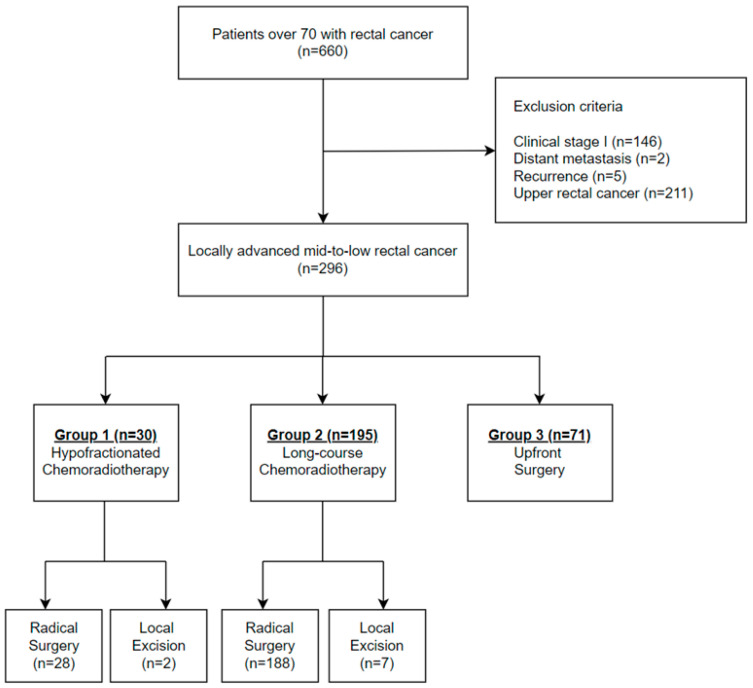
Flow diagram of the study.

**Figure 2 cancers-16-04280-f002:**
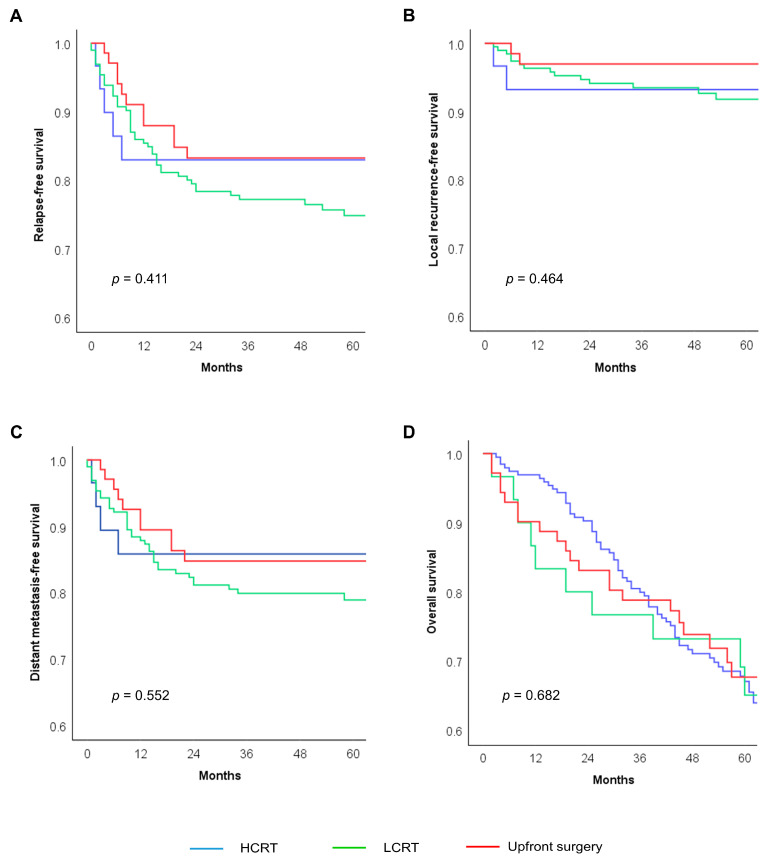
Kaplan–Meier curves demonstrating (**A**) relapse-free survival, (**B**) local recurrence-free survival, (**C**) distant metastasis-free survival, and (**D**) overall survival. HCRT, hypofractionated chemoradiotherapy; LCRT, long-course chemoradiotherapy.

**Table 1 cancers-16-04280-t001:** Baseline characteristics.

Characteristics		HCRT (n = 30)	LCRT (n = 195)	Upfront Surgery (n = 71)	*p* Value
Sex	Male	15 (50.0)	124 (63.6)	34 (47.9)	0.303
	Female	15 (50.0)	71 (36.4)	37 (52.1)	
Age (years)		78.0 ± 5.1	76.3 ± 4.4	76.9 ± 4.5	0.129
ASA score	1, 2	17 (56.7)	162 (83.1)	60 (84.5)	0.002
	3, 4	13 (43.3)	33 (16.9)	11 (15.5)	
BMI (kg/m^2^)	<25	19 (63.3)	139 (71.3)	59 (83.1)	0.067
	≥25	11 (36.7)	56 (28.7)	12 (16.9)	
Albumin (g/dL)	<3.5	1 (3.3)	17 (8.7)	9 (12.7)	0.311
	≥3.5	29 (96.7)	178 (91.3)	62 (87.3)	
Hemoglobin (g/dL)	<12	14 (46.7)	97 (49.7)	26 (36.6)	0.165
	≥12	16 (53.3)	98 (50.3)	45 (63.4)	
Pretreatment CEA (ng/mL)	<5	14 (46.7)	103 (52.8)	47 (66.2)	0.091
	≥5	16 (53.3)	92 (47.2)	24 (33.8)	
Tumor location	Mid	18 (60.0)	107 (54.9)	61 (85.9)	<0.001
	Low	12 (40.0)	88 (45.1)	10 (14.1)	
Tumor differentiation	WD	6 (20.0)	80 (41.0)	29 (40.8)	0.135
	MD	21 (70.0)	108 (55.4)	40 (56.3)	
	PD, mucinous	3 (10.0)	7 (3.6)	2 (2.8)	
Clinical T stage	1, 2	1 (3.3)	4 (2.1)	3 (4.2)	0.277
	3	27 (90.0)	161 (82.6)	63 (88.7)	
	4	2 (6.7)	30 (15.4)	5 (7.0)	
Clinical N stage	0	18 (60.0)	94 (48.2)	32 (45.1)	0.459
	1	8 (26.7)	61 (31.3)	28 (39.4)	
	2	4 (13.3)	40 (20.5)	11 (15.5)	

Data are presented as mean ± standard deviation or number (percentage). HCRT, hypofractionated chemoradiotherapy; LCRT, long-course chemoradiotherapy; ASA, American Society of Anesthesiologists; BMI, body mass index; CEA, carcinoembryonic antigen; WD, well differentiated; MD, moderately differentiated; PD, poorly differentiated.

**Table 2 cancers-16-04280-t002:** Radiotherapy-related complications.

	HCRT (n = 30)	LCRT (n = 195)	*p* Value
Complications			0.001
Yes	5 (16.7)	95 (48.7)	
No	25 (83.3)	100 (51.3)	
Types			
Nausea/vomiting	1 (3.3)	8 (4.1)	
Headache	0 (0.0)	9 (4.6)	
Bowel habit change	2 (6.7)	38 (19.5)	
Anorexia	2 (6.7)	22 (11.3)	
Dermatitis	0 (0.0)	10 (5.1)	
Dysuria	1 (3.3)	16 (8.2)	
Anal pain	2 (6.7)	43 (22.1)	
Abdominal pain	1 (3.3)	9 (4.6)	
General weakness	0 (0.0)	5 (2.6)	
Leg edema	0 (0.0)	1 (0.5)	
CTCAE grade			0.911
1	4 (13.3)	76 (38.9)	
2	1 (3.3)	16 (8.2)	
3, 4	0 (0.0)	3 (1.0)	

The numbers were counted repeatedly if the patient experienced more than one complication. CTCAE, Common Terminology Criteria for Adverse Events.

**Table 3 cancers-16-04280-t003:** Pathological characteristics in patients with neoadjuvant chemoradiotherapy.

Characteristic		HCRT (n = 30)	LCRT (n = 195)	*p* Value
pCR	Yes	3 (10.0)	30 (15.4)	0.002
	No	27 (90.0)	165 (84.6)	
Pathological T stage	0, 1	3 (10.0)	42 (21.5)	0.243
	2	7 (23.3)	39 (20.0)	
	3	16 (53.3)	103 (52.8)	
	4	4 (13.3)	11 (5.6)	
Pathological N stage	0	18 (64.3)	125 (66.5)	0.059
	1	5 (17.9)	52 (27.7)	
	2	5 (17.9)	11 (5.9)	

Data are presented as number (percent). HCRT, hypofractionated chemoradiotherapy; LCRT, long-course chemoradiotherapy; pCR, pathological complete response.

**Table 4 cancers-16-04280-t004:** Univariate analysis for survival outcomes.

Variables		n	Relapse-Free Survival		Local Recurrence-Free Survival		Overall Survival	
			3-Year RFS (%)	*p*	3-Year LRFS (%)	*p*	5-Year OS (%)	*p*
Sex	Male	173	77.6	0.555	94.4	0.889	64.0	0.128
	Female	123	82.3		95.6		71.1	
Age (years)	<80	219	80.2	0.531	94.3	0.887	71.6	0.007
	≥80	77	77.6		96.8		53.2	
ASA score	1, 2	239	79.3	0.852	93.7	0.162	68.3	0.150
	3, 4	57	80.9		100.0		60.7	
BMI (kg/m^2^)	<25	217	81.6	0.379	95.1	0.685	65.9	0.872
	≥25	79	74.2		94.4		70.1	
Albumin (g/dL)	<3.5	27	87.0	0.655	100.0	0.719	43.0	0.022
	≥3.5	269	79.1		94.5		69.2	
Hemoglobin (g/dL)	<12	137	71.6	0.001	91.8	0.012	53.9	<0.001
	≥12	159	86.3		97.4		77.9	
Pretreatment CEA (ng/mL)	<5	164	84.5	0.007	95.4	0.301	69.7	0.239
	≥5	132	73.5		94.2		63.4	
Tumor location	Mid	186	83.6	0.010	98.3	<0.001	69.1	0.039
	Lower	110	72.9		89.0		63.2	
Tumor differentiation	WD	114	87.2	<0.001	96.2	<0.001	70.5	0.003
	MD	168	76.6		95.5		67.6	
	PD, mucinous	9	42.9		71.4		33.3	
Clinical T stage	1, 2	8	87.5	0.019	87.5	0.016	71.4	0.648
	3	251	81.8		96.5		68.4	
	4	37	63.7		86.0		56.8	
Clinical N stage	0	144	89.1	<0.001	97.7	0.047	74.5	0.014
	1	97	73.3		94.2		66.6	
	2	55	65.9		88.5		47.4	
Adjuvant chemotherapy	(−)	106	80.3	0.849	95.5	0.511	49.5	<0.001
	(+)	190	79.2		94.6		76.6	
CRM	(−)	137	87.0	<0.001	96.9	0.207	72.3	0.044
	(+)	44	58.8		90.0		53.5	
LVI	(−)	256	83.2	<0.001	95.3	0.004	62.1	0.001
	(+)	40	52.5		85.0		37.5	
PNI	(−)	224	84.8	<0.001	96.4	0.001	62.1	<0.001
	(+)	72	61.1		86.1		48.6	
TRG	≥2	35	68.6	0.049	85.7	0.049	31.4	<0.001
	3	154	76.6		92.9		59.7	
	4	33	90.9		100.0		75.8	
Treatment strategy	HCRT	30	83.0	0.411	93.1	0.464	65.1	0.682
	LCRT	195	77.2		93.2		67.0	
	Upfront surgery	71	83.2		93.5		67.7	

RFS, relapse-free survival; LRFS, local recurrence-free survival; OS, overall survival; ASA, American Society of Anesthesiologists; BMI, body mass index; CEA, carcinoembryonic antigen; CRM, circumferential resection margin; LVI, lymphovascular invasion; PNI, perineural invasion; TRG, tumor regression grade; HR, hazard ratio; CI, confidence interval.

**Table 5 cancers-16-04280-t005:** Multivariate analysis for survival outcomes.

Variables	Relapse-Free Survival ^(a)^			Local Recurrence-Free Survival ^(b)^			Overall Survival ^(c)^		
	HR	95% CI	*p*	HR	95% CI	*p*	HR	95% CI	*p*
Treatment strategy									
HCRT vs. LCRT	0.875	0.334–2.291	0.785	1.385	0.276–6.947	0.692	0.962	0.491–1.888	0.911
Surgery vs. LCRT	0.650	0.315–1.340	0.243	0.381	0.067–2.172	0.277	0.926	0.560–1.532	0.765

HR, hazard ratio; CI, confidence interval; HCRT, hypofractionated chemoradiotherapy; LCRT, long-course chemoradiotherapy. ^(a)^ Included covariables were hemoglobin, pretreatment CEA, tumor location, tumor differentiation, clinical T stage, clinical N stage, and treatment strategy. ^(b)^ Included covariables were hemoglobin, tumor location, tumor differentiation, clinical T stage, clinical N stage, and treatment strategy. ^(c)^ Included covariables were age, albumin, hemoglobin, tumor location, tumor differentiation, clinical N stage, adjuvant chemotherapy, and treatment strategy. Tumor regression grade (TRG) was excluded as a covariate due to the unavailability of data in the upfront surgery group.

## Data Availability

The data presented in this study are available on request from the corresponding author. The data are not publicly available due to personal data protection.

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
