# Peer review of "Efficacy of Neoadjuvant Hypofractionated Chemoradiotherapy in Elderly Patients with Locally Advanced Rectal Cancer: A Single-Center Retrospective Analysis"

_cancers, 2024, doi:10.3390/cancers16244280_

Round 1
Reviewer 1 Report
Comments and Suggestions for Authors
Manuscript entitled "Efficacy of Neoadjuvant Hypofractionated Chemoradiotherapy in Elderly Patients with Locally Advanced Rectal Cancer: A Single-center Retrospective Analysis"
Major issues:
1. The authors should list some important clinical pathologic variables in correlation and survival analysis, including lymphovascular and perineurial invasions.
2. The authors should provide tumor regression grade in correlation and survival analysis.
3. The authors should provide molecular status (KRAS, BRAF, MMR status) in correlation and survival analysis.
4. The distal metastasis-free survival should also be analyzed.
Reviewer 2 Report
Comments and Suggestions for Authors
The purpose of this study was to evaluate the efficacy of neoadjuvant hypofractionated chemoradiotherapy (HCRT), a new treatment modality for locally advanced rectal cancer in the elderly. The study compared the advantages of HCRT over conventional long-term chemoradiotherapy (LCRT) and surgery in terms of long-term survival and risk of complications.
This study is very interesting. However, there are several concerns that should be discussed and are presented below.
What is the specific method of HCRT used in this study?
Have similar results been obtained in other studies?
How will the results of this study translate to clinical practice?
I would like to know more about the side effects of HCRT.
How do the results of this study apply to other types of tumors?
Can they be applied to elderly patients with sarcoma? If possible, please discuss the following references. Clinical outcomes of patients with primary malignant bone and soft tissue tumor aged 65 years or older. Exp Ther Med. 2019 Jan;17(1):888-894. doi: 10.3892/etm.2018.7013. Epub 2018 Nov 26. PMID: 30651877; PMCID: PMC6307412.
Reviewer 3 Report
Comments and Suggestions for Authors
This paper examines the effectiveness of neoadjuvant hypofractionated chemoradiotherapy in elderly patients with locally advanced rectal cancer. It compares HCRT with long-course chemoradiotherapy and upfront surgery, analyzing treatment outcomes, complications, and survival rates in patients aged 70 years or older with mid-to-low LARC.
My comments:
Method:
It is essential to include a complete statement regarding the IRB approval for this study. This statement should explicitly mention adherence to the principles outlined in the Declaration of Helsinki, which is fundamental for ensuring ethical standards in medical research involving human subjects. Moreover, the authors should provide clear information about the process of obtaining informed consent from the patients or their legal representatives.
Sample Size: The HCRT group (n=30) is considerably smaller than the LCRT group (n=195). This disparity may affect the statistical power of the comparisons. A power analysis would be beneficial to determine if the sample size is adequate to draw meaningful conclusions.
The criteria for selecting patients for each treatment modality are not clearly defined. It would be helpful to understand if there were specific reasons why certain patients received HCRT over LCRT or upfront surgery.
More information on the radiotherapy planning and delivery techniques.
Discussion:
The discussion section could be expanded to provide more specific guidance on patient selection for HCRT based on the study's findings.
Round 2
Reviewer 1 Report
Comments and Suggestions for Authors
The revision is acceptable for publication.
Reviewer 2 Report
Comments and Suggestions for Authors
The authors replied well, so the manuscript is suitable for publication.
Reviewer 3 Report
Comments and Suggestions for Authors
The paper is ready for publication.